# Rural-Urban differentials in prevalence, spectrum and determinants of Non-alcoholic Fatty Liver Disease in North Indian population

Md Asadullah[1], Roopa Shivashankar[2], Shalimar[3], Devasenathipathy Kandasamy[4], Dimple Kondal[5], Garima Rautela[2], Ariba Peerzada[6], Bhanvi Grover[6], Ritvik Amarchand[1], Baibaswata Nayak[3], Raju Sharma[4], Lakshmy Ramakrishnan[7], Dorairaj Prabhakaran[2], Anand Krishnan[1], Nikhil Tandon[6]*

1 Centre for Community Medicine, All India Institute of Medical Sciences, New Delhi, India, 2 Centre for Chronic Disease Control, New Delhi, India, 3 Department of Gastroenterology, All India Institute of Medical Sciences, New Delhi, India, 4 Department of Radio-diagnosis, All India Institute of Medical Sciences, New Delhi, India, 5 Public Health Foundation of India, Gurgaon, India, 6 Department of Endocrinology and Metabolism, All India Institute of Medical Sciences, New Delhi, India, 7 Department of Cardiac Biochemistry, All India Institute of Medical Sciences, New Delhi, India

* nikhil_tandon@hotmail.com

**Data Availability Statement:** The data can't be shared publicly because the consent for data sharing was not obtained from study participants.

## Abstract

### Background

Non-alcoholic fatty liver disease (NAFLD) is a spectrum of disease ranging from simple steatosis, non-alcoholic steatohepatitis (NASH), through to advanced fibrosis and cirrhosis. We assessed the prevalence, spectrum, and determinants of NAFLD among adults in urban and rural North India.

### Methods

A representative sample of adults aged 30–60 years were recruited from urban Delhi and rural Ballabhgarh during 2017–2019. Participants underwent abdominal ultrasonography (USG) and vibration controlled transient elastography (VCTE) with FibroScan to assess fatty liver and fibrosis, respectively. We estimated the age- and sex-standardised prevalence of NAFLD and its spectrum. The factors associated with 'ultrasound-diagnosed NAFLD' were identified using multivariate logistic regression.

### Results

A total of 828 urban (mean ± SD age: 45.5 ± 8.0 years; women: 52.7%) and 832 rural (mean ± SD age: 45.1 ± 7.9 years; women: 62.4%) participants were recruited. The age- and sex-standardized prevalence of ultrasound-diagnosed NAFLD was 65.7% (95%CI: 60.3–71.2) in the urban and 61.1% (55.8–66.5) in the rural areas, respectively. The prevalence of NAFLD with elevated alanine transaminase (≥40IU/L) was 23.2% (19.8–26.6), and 22.5% (19.0–26.0) and any fibrosis by liver stiffness measurement on transient elastography (≥6.9 kPa) was 16.5% (13.8–19.8) and 5.2% (3.8–6.7) in urban and rural participants,

The data set used in this study are available at the Research Section of the All India Institute of Medical Sciences, New Delhi. The researchers meeting the criteria can access the data by writing an email to "deanresearchaiims@gmail.com" with the subject line "Request for data for the manuscript Rural-Urban differentials in prevalence, spectrum and determinants of Non-alcoholic Fatty Liver Disease in North Indian population (Author: Tandon)".

**Funding:** This study was financially supported by Indian Council of Medical Research (ICMR), New Delhi, India (Grant No. 5/4/3-3/TF/2012/NCD-II) . Funding agency had no role in study design, data collection, analysis, decision to publish, or preparation of the manuscript.

**Competing interests:** The authors have declared that no competing interests exist.

respectively. In both urban and rural areas, diabetes, central obesity and insulin resistance were significantly associated with NAFLD.

## Conclusion

NAFLD prevalence was high among rural and urban North Indian adults, including fibrosis or raised hepatic enzymes. The strong association of metabolic determinants confirms its linkage with metabolic syndrome.

## Introduction

Non-alcoholic fatty liver disease (NAFLD) is the most common chronic liver disease worldwide [1]. The clinical spectrum of NAFLD ranges from simple steatosis, steatohepatitis (NASH), hepatic fibrosis, cirrhosis to end-stage liver disease with related complications, which includes hepatocellular carcinoma (HCC) [2]. NAFLD, represents the hepatic component of the metabolic syndrome (MS) and is associated with other MS components—obesity, insulin resistance, type 2 diabetes mellitus, dyslipidemia, hypertension, and cardiovascular diseases [3, 4]. The estimated global prevalence of NAFLD in adults older than 18 years of age is 25.2%, with the highest levels reported in the Middle-East and South America, followed by Asia, North America, Europe and Africa [5]. A recent meta-analysis reported a significant increase in the prevalence of NAFLD in Asia over time. It indicated a 29.6% pooled prevalence in Asia with wide variation across countries [1].

The public health importance of NAFLD stems from its impact on morbidity, mortality and health care utilization globally. NASH and hepatic fibrosis are associated with excess all-cause mortality, cardiovascular mortality, and liver-related mortality in the general population [6, 7]. NASH is one of the leading causes of chronic liver disease and cirrhosis of the liver. It is the fastest emerging indication for liver transplantation in the United States, United Kingdom and many low and middle-income countries [8]. The excess cardiovascular risk in fatty liver has been reported even in populations like India, with relatively low body mass index (BMI) [9].

A recently published systematic review and meta-analyses of the prevalence of MS among India's adult population based on 111 studies estimated its prevalence to be 30% (95%CI: 28%-33%). The authors reported significantly higher prevalence with increasing age, in facility-based studies, in urban areas, in women but did not find any significant regional or temporal (2003 to 2019) trends [10]. However, there are very few studies on NAFLD from India. Most studies included specific groups like obese or non-obese, people with diabetes, women with gestational diabetes etc [11–15]. Few population-based studies were available at the time of planning of the present study reported a prevalence of NAFLD between 19% to 32.0% [16–18]. Subsequently, another population study from Kerala reported a high prevalence from urban (55.2%) and rural (43.4%) populations evaluated concurrently [19]. Almost all the studies used ultrasound to diagnose NAFLD and did not assess the full spectrum of NAFLD.

To accurately measure the full burden due to NAFLD in India, there is a need to cover its entire spectrum, not just based on the gradation of steatosis but also in terms of hepatic injury measured by raised alanine aminotransferase (ALT) and fibrosis. In this study, we aimed to assess the prevalence, spectrum, and metabolic determinants of NAFLD among adults in urban and rural North India.

## Participants and methods

### Study population and design

We undertook a community-based cross-sectional study of representative adults aged between 30 to 60 years in urban Delhi and rural Ballabhgarh block of Faridabad district of Haryana State during December 2017- April 2019. We estimated a sample size of 825 for both urban and rural areas using a design effect of 1.5, an alpha error of 5% and a 30% non-response rate. Based on the available data at the time of conception, we assumed a prevalence of 20% and 10% with absolute margins of errors of 4% and 3% for urban and rural areas, respectively. [11, 16].

The Institutional Ethics Committees of the All India Institute of Medical Sciences (AIIMS) and Centre for Chronic Disease Control (CCDC), New Delhi, approved the study.

### Sampling strategy

Both the rural and urban components of the study used a previously available community-level sampling frame. The urban Delhi participants were recruited from the ongoing Centre for Cardiometabolic Risk Reduction in South Asia (CARRS) cohort study, which recruited adults aged $\geq$ 20 years in 2010–11 using a multistage cluster random sampling [20]. The participants from rural Ballabhgarh were recruited from an earlier ICMR funded Coronary Heart Disease (CHD) repeat survey in 2010–11 among adults aged $\geq$ 30 years selected through a random sampling of households [21]. The youngest participants, ICMR-CHD repeat survey participants at the beginning of NAFLD study were 38 years. Therefore, to compensate for the study participants' age difference, an additional 136 subjects aged 30 to 37 years living in the same households were recruited in the rural arm. A written informed consent was obtained from all participants. All identified subjects were screened using a questionnaire. Based on this, we excluded individuals who reported alcohol consumption of any type equivalent to >140gm/ week (>20gm/day) [4]. Other exclusions included history of cirrhosis of the liver, hepatocellular carcinoma (HCC), those who were positive for antibodies for hepatitis B virus and hepatitis C virus. Bedridden individuals and pregnant women were also excluded.

### Data collection

The recruited urban participants were invited to the AIIMS, Delhi, and the rural participants to Qimat Rai Gupta (QRG) Health City Hospital, Faridabad, for data collection. Trained interviewers administered the questionnaire in Hindi to elicit information on the socio-economic and demographic details, detailed history of tobacco and alcohol use, diet (food frequency questionnaire), physical activity (Global Physical Activity Questionnaire [GPAQ]), history of cardiometabolic diseases (CMDs), liver diseases, medication use and quality of life. All data were collected electronically using Samsung Galaxy tablets (Model no: SM-T355Y).

All participants underwent anthropometric measurements- weight, standing height, waist and hip circumferences using standard techniques. Blood pressure (BP) was measured two times using a validated electronic BP measuring device (Omron HEM 7080). A third reading of blood pressure was obtained if the difference between the first two readings of systolic blood pressure was $\geq$10 mm of Hg or/and diastolic blood pressure was $\geq$ 6 mm of Hg. The mean of the last two readings was considered for analysis.

We collected 18 ml fasting blood samples and estimated the fasting plasma glucose (FPG), glycosylated haemoglobin (HbA1c), serum insulin, lipids (total-, very low, low and high-density lipoprotein [VLDL, LDL and HDL] cholesterol and triglycerides), liver enzymes (alanine aminotransferase (ALT), aspartate aminotransferase (AST), alkaline phosphatase (ALP)),

> ## Box 1. Definitions used in the study
>
> **NAFLD** was defined and graded on ultrasound as mild (Grade 1), moderate (Grade 2) and severe (Grade 3) based on visual analysis of the intensity of echogenicity of the liver.
>
> **NAFLD** was defined based on a CAP cut off value of $\geq$275 dB/m for BMI<30 kg/m$^2$ or $\geq$285 dB/m for BMI$\geq$30 Kg/m$^2$.
>
> **Raised Alanine Transaminase** (ALT) defined as levels $\geq$ 40 IU.
>
> **Presence of any hepatic fibrosis** ($\geq$F1), significant fibrosis ($\geq$F2), advanced fibrosis ($\geq$F3) and cirrhosis (F4) was based on liver stiffness measurement (LSM) cut off values of 6.9 kPa, 7.9 kPa, 8.6 kPa, and 14.2 kPa respectively.
>
> **Central obesity** was defined as the presence of waist circumference $\geq$90 cm for men or $\geq$80 cm for women.
>
> **Diabetes** was defined as fasting plasma glucose $\geq$126 mg/dl and/or HbA1c $\geq$6.5% and/or reported being on anti-diabetes medication.
>
> **Hypertension** was defined as having a systolic blood pressure $\geq$140 or/and diastolic blood pressure $\geq$90 mmHg or reported being on anti-hypertensive medication.
>
> Source [22–26].

hepatic viral markers (HbsAg and Anti-HCV) and biomarkers of kidney function (serum urea, and creatinine).

A trained medical professional did abdominal ultrasonography (USG) using a 3.5–5 MHz curvilinear transducer (Siemens–Model: ACUSON X300 in urban and PHILIPS-Model: IU22 G Cart in rural). The images were stored, read and graded by a Radiologist (DK) of AIIMS, New Delhi. Ten percent of blinded ultrasound images were re-read by DK to document the intra- person agreement in grading scans as normal, Grade 1, 2 and 3 NAFLD. The kappa value for intra-person agreement for grading the scans was 0.66 (0.55–0.76) indicating reasonable agreement between the two readings.

All participants underwent vibration controlled transient elastography (VCTE)- with FibroScan 502 Touch (ECHOSENS, Paris, France) by a trained technician using M probe in participants with BMI <30 Kg/m$^2$ and XL probe in participants with BMI $\geq$30 Kg/m$^2$ for the measurement of the controlled attenuated parameter (CAP) and liver stiffness measurement (LSM). The definitions used in the study are given in Box 1.

### Statistical analysis

We excluded participants positive to hepatic viral markers or with non-readable ultrasound images from the dataset. We tabulated the population's baseline characteristics–categorical data were presented as proportions (95% Confidence intervals), and continuous data were reported as mean ± standard deviation (SD). Differences between the two groups were assessed by chi-square test for categorical data and t-test for continuous data. We stratified age by decades and used standard stratifications for socio-economic indicators (education and income).

To better compare the rural and urban prevalence, given the differences in age and sex structure of the rural and urban sample, we calculated standardized prevalence using the

Indian Census 2011 data as the standard population [27]. For analysis, we used multiple criteria to define NAFLD- based on ultrasound, ultrasound with ALT, and CAP values. The definitions of each criterion are shown in Box 1. All the analyses were done separately for the urban and rural populations. Finally, we performed multiple logistic regression to measure the strength of association between diabetes, hypertension, central obesity and raised lipids with NAFLD. We did separate models for urban and rural areas and the combined data (with additional adjusting for residence). All the models were adjusted for age and sex. Statistical analyses were performed using STATA statistical software version 15.0.

## Results

From a sampling frame of 1120 and 1837, we recruited 855 participants in urban and 876 subjects in rural areas, respectively (Fig 1). After excluding HBsAg or anti-HCV antibody positive individuals and those whose ultrasound images were non-readable, we were left with 828 urban (women: 52.7%) and 832 rural (women: 62.4%) participants for the final analysis.

The background characteristics of the subjects are shown in Table 1. The mean (±SD) age of urban (45.5 ± 8.0 years) and rural participants (45.1 ± 7.9 years) was similar. Compared to the urban population, the rural cohort had a higher prevalence of illiteracy (32.5% vs 9.8%) and a monthly family income of less than ≤ INR 10,000 (20.0% vs 13.3%). BMI and waist circumference were significantly higher in urban subjects as compared to rural subjects (p<0.001). Among biochemical variables, the mean levels of total cholesterol, the proportion with lower HDL cholesterol (<40 mg/dl for men and <50 mg/dl for women), insulin resistance and diabetes were significantly worse in urban areas (p<0.001). The prevalence of hypertension was relatively higher in urban (31.3%) than rural (26.9%) subjects (p = 0.051).

Table 2 provides the age and sex standardised prevalence of NAFLD in the urban, rural and total population. The age and sex standardized NAFLD (based on definitions described in Box 1) prevalence diagnosed by ultrasound did not show urban -rural (65.7% vs 61.1%) or gender difference (men vs women: 64.2% vs 67.4% in urban and 63.6% vs 58.6% in rural). NAFLD diagnosed by CAP score in VCTE showed a much higher prevalence in urban areas (53.7%; 95%CI: 48.8–58.5) than rural areas (33.7%; 95%CI: 29.7–37.7). The prevalence of ultrasound

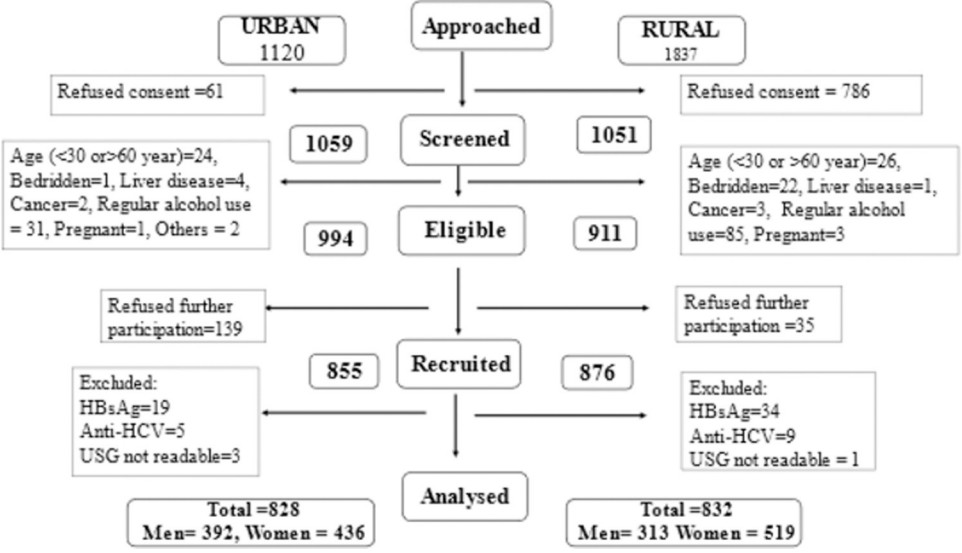

**Fig 1. Flow chart showing recruitment of the participants in the study.**

**Table 1. Characteristics of study participants.**

| Indicators | Urban (n = 828) | Rural (n = 832) | P-value |
|---|---|---|---|
| **Socio-demographic characteristics** | | | |
| Mean (±SD) Age in years | 45.5 ± 8.0 | 45.1 ± 7.9 | 0.315 |
| Age Category | | | 0.246 |
| 30–39 Year | 26.0% | 26.7% | |
| 40–49 Year | 38.3% | 41.3% | |
| 50–60 Year | 35.7% | 32.0% | |
| % Women | 52.7% | 62.4% | <0.001 |
| % Illiterate | 9.8% | 32.5% | <0.001 |
| Monthly family income ≤10,000 INR | 13.3% | 20.0% | <0.001 |
| **Metabolic characteristics** | | | |
| Mean (±SD) body mass index (Kg/m$^2$)* | 27.2 ± 4.9 | 25.3 ± 4.7 | <0.001 |
| Mean (±SD) waist circumference (cm)# | 92.6 ± 11.7 | 89.7 ± 12.2 | <0.001 |
| Central Obesity#@ | 73.6% | 70.9% | 0.215 |
| General obesity (BMI ≥25.0 Kg/m$^2$)* | 65.2% | 49.9% | <0.001 |
| Overweight (BMI ≥23.0 Kg/m$^2$)* | 16.3% | 19.9% | |
| Raised total cholesterol (≥200 mg/dl) | 28.4% | 43.3% | <0.001 |
| Elevated triglyceride (≥150 mg/dl) | 39.5% | 40.6% | 0.638 |
| Reduced HDLc^ | 64.7% | 49.3% | <0.001 |
| Insulin Resistance$ | 63.9% | 43.8% | <0.001 |
| Prediabetes** | 66.6% | 65.5% | 0.654 |
| Diabetes## | 30.1% | 14.8% | <0.001 |
| Hypertension@@ | 31.3% | 26.9% | 0.051 |

*Weight and height were taken for 824 subjects in urban and 831 in rural area.

# waist circumference was measured in 827 urban subjects and 832 rural subjects.

@Central obesity—waist circumference ≥90 cm for men or ≥80 cm for women.

^Reduced HDLc: HDL cholesterol<40 mg/dl in men or <50 mg/dl in women.

$Insulin Resistance: HOMA-IR: Fasting insulin(μIU/ml)*fasting glucose(mmol/L)/22.5, Cut off value > = 2.5.

**Pre-diabetes: no prior diagnosis of diabetes and (FPG 100–125 mg/dl and/or HbA1c 5.7–6.4%).

##Diabetes: Fasting plasma glucose ≥126 mg/dl and/ or HbA1c ≥6.5% and/or reported being on anti-diabetes medication.

@@Hypertension: Systolic blood pressure ≥ 140 mmHg or diastolic blood pressure ≥ 90 mmHg or on antihypertensive drug for hypertension.

diagnosed NAFLD with raised ALT levels was 23.2% in urban and 22.5% in rural subjects with a clear male preponderance. The urban and rural prevalence of hepatic fibrosis was 16.5% vs 5.2%, and cirrhosis was 2.8% vs 0.6%, respectively. The crude prevalence of NAFLD's spectrum is shown in S1 Table.

**Table 2. Standardized* prevalence (%; 95%CI) of spectrum# of NAFLD in the study population.**

| | Urban | | | Rural | | |
|---|---|---|---|---|---|---|
| **Spectrum** | **Men** | **Women** | **Both** | **Men** | **Women** | **Both** |
| NAFLD based on Ultrasound | 64.2 (56.2–72.1) | 67.4 (59.8–75.0) | 65.7 (60.3–71.2) | 63.6 (54.5–72.7) | 58.6 (52.1–65.1) | 61.1 (55.8–66.5) |
| NAFLD based on VCTE | 56.8 (49.5–64.1) | 50.4 (43.9–56.9) | 53.7 (48.8–58.5) | 38.3 (31.3–45.4) | 28.8 (24.4–33.3) | 33.7 (29.7–37.7) |
| NAFLD on USG with ALT≥40 IU/L | 33.2 (27.4–39.0) | 12.8 (9.5–16.1) | 23.2 (19.8–26.6) | 33.4 (26.7–40.1) | 11.2 (8.4–14.0) | 22.5 (19.0–26.0) |
| Fibrosis (≥F1) based on LSM value | 17.8 (13.9, 21.8) | 15.0 (11.5, 18.6) | 16.5 (13.8, 19.8) | 6.4 (3.7, 9.0) | 4.1 (2.5, 5.6) | 5.2 (3.8, 6.7) |
| F4 based on LSM value (Cirrhosis) | 2.7 (1.2, 4.2) | 3.0 (1.4, 4.6) | 2.8 (1.7, 4.0) | 1.0 (-0.1, 2.0) | 0.2 (-0.2, 0.6) | 0.6 (0.0, 1.1) |

*Standardized to age and sex structure of Indian population census 2011.

#for all definitions see Box 1.

**Table 3. Presence of raised alanine transaminase (ALT) levels and any fibrosis according to the severity of fatty liver assessed by ultrasound.**

| Parameter | | Grade of fatty liver on ultrasound | | | | P—value |
|---|---|---|---|---|---|---|
| | | Normal (%) | Grade 1 (%) | Grade 2 (%) | Grade 3 (%) | |
| Urban | N | 273 | 358 | 153 | 44 | |
| | ALT ≥40 IU/L | 13.6 | 22.1 | 52.9 | 52.3 | <0.001 |
| | Any fibrosis# | 10.6 | 14.3 | 31.4 | 43.2 | <0.001 |
| Rural | N | 330 | 402 | 95 | 5 | |
| | ALT ≥40 IU/L | 12.7 | 23.4 | 62.1 | 60.0 | <0.001 |
| | Any fibrosis# | 2.7 | 5.0 | 17.9 | 40.0 | <0.001 |

# Controlled Attenuation Parameter (CAP) ≥275 dB/m for BMI<30 kg/m2 or CAP ≥285 dB/m for BMI≥30 Kg/m$^2$.

The agreement between ultrasound and CAP score as modalities for assessing fatty liver, as measured by kappa value, was fair [28] in both urban (0.39; 0.33–0.46) and rural (0.31; 0.26–0.37) samples. Almost 95% of those diagnosed as NAFLD by ultrasound and deemed normal by CAP score were rated as Grade 1 NAFLD (data not shown). The proportion of subjects with raised ALT levels and fibrosis based on LSM values increased with the grade of fatty liver (measured by ultrasound). About 13% of both urban and rural subjects with no ultrasound detected NAFLD had raised ALT levels. Fibrosis (≥F1) assessed by LSM values among those with normal liver on ultrasound was 10.6% in urban and 2.7% in rural subjects (Table 3).

Multivariate logistic regression identified presence of diabetes (OR: 2.06; 95%CI: 1.34–3.17); overweight (OR: 2.06; 95% CI: 1.19–3.54), central obesity (OR: 1.79; 95%CI: 1.07–2.98) and insulin resistance (OR: 2.30; 95%CI: 1.58–3.34) as significant determinants of ultrasound proven NAFLD in urban areas. In the rural area, the significant determinants identified were diabetes (OR: 2.11; 95%CI: 1.22–3.66), central obesity (OR: 2.29; 95%CI: 1.42–3.69), and insulin resistance (OR: 1.63; 95%CI: 1.13–2.33) (Table 4). Higher age (45–60 years) appeared to be protective (OR: 0.69; 95%CI: 0.48–0.98) in the urban area.

NAFLD prevalence increased with age in both urban and rural women. By contrast, the prevalence of NAFLD decreased with age among men in both urban and rural areas (Fig 2).

**Table 4. Determinants of NAFLD (defined on ultrasound) in urban and rural subjects.**

| Factors | Urban | | Rural | |
|---|---|---|---|---|
| | Univariate | Multivariate | Univariate | Multivariate |
| | OR (95%CI) | OR (95%CI) | OR (95%CI) | OR (95%CI) |
| Age 45–60 yrs | 1.03 (0.77–1.38) | 0.69 (0.48–0.98) | 1.07 (0.81–1.41) | 1.01 (0.73–1.41) |
| Women | 1.24 (0.93–1.66) | 0.90 (0.63–1.29) | 1.04 (0.78–1.38) | 0.91 (0.64–1.30) |
| Diabetes | 3.16 (2.18–4.56) | 2.06 (1.34–3.17) | 3.75 (2.29–6.14) | 2.11 (1.22–3.66) |
| Hypertension | 1.46 (1.06–2.01) | 0.96 (0.66–1.40) | 1.79 (1.29–2.48) | 1.11 (0.76–1.63) |
| Overweight | 2.66 (1.62–4.36) | 2.06 (1.19–3.54 | 1.99 (1.30–3.02) | 1.28 (0.79–2.07) |
| Central Obesity | 4.68 (3.37–6.50) | 1.79 (1.07–2.98) | 5.52 (3.99–7.64) | 2.29 (1.42–3.69) |
| Total cholesterol ≥s200 mg/dl | 1.33 (0.96–1.86) | 1.01 (0.67–1.53) | 1.78 (1.34–2.37) | 1.34 (0.93–1.92) |
| Triglycerides ≥150 mg/dl | 2.19 (1.60–3.01) | 1.41 (0.97–2.04) | 1.86 (1.39–2.48) | 1.17 (0.82–1.66) |
| HOMA-IR | 4.80 (3.52–6.55) | 2.30 (1.58–3.34) | 3.52 (2.60–4.76) | 1.63 (1.13–2.33) |
| Reduced HDL | 1.35 (1.00–1.82) | 0.94 (0.64–1.38) | 1.19 (0.90–1.57) | 1.07 (0.76–1.51) |

Diabetes: Fasting plasma glucose≥126 mg/dl and/ or HbA1c ≥6.5% and/or reported being on anti-diabetes medication; Hypertension: Systolic blood pressure ≥ 140 mmHg or diastolic blood pressure ≥ 90 mmHg or on antihypertensive drug for hypertension; Central obesity: waist circumference ≥90 cm for Asian men or ≥80 cm for Asian women; Elevated TC: Reduced HDLc: HDL cholesterol<40 mg/dl in men or <50 mg/dl in women.

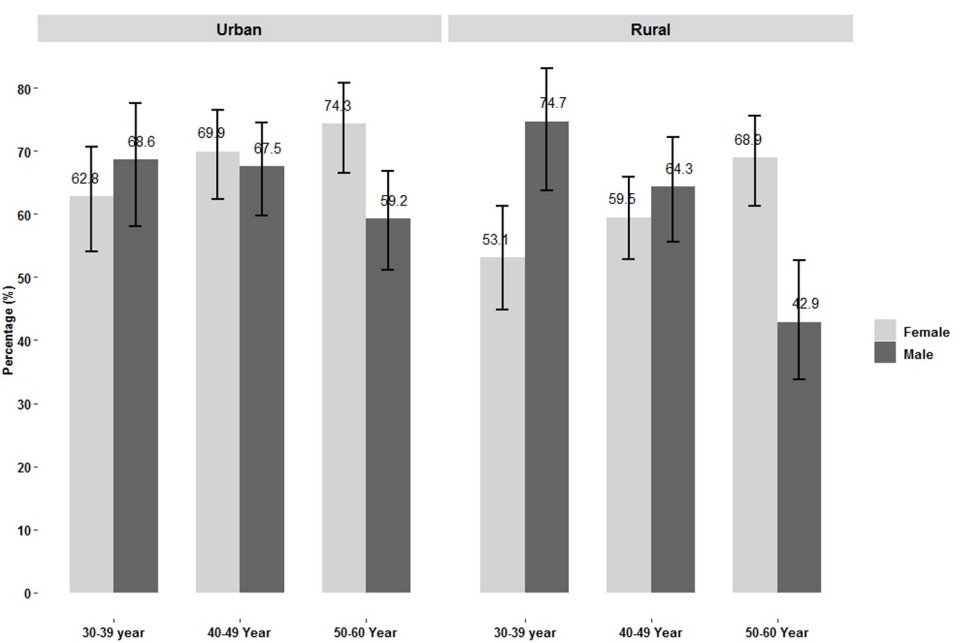

**Fig 2. Prevalence of NAFLD on ultrasound by age and sex among urban and rural study subjects.**

When we did a combined multivariate analysis including both urban and rural data, we found that the place of residence (urban vs rural) was not a significant determinant of the prevalence of NAFLD (OR: 1.04; 95%CI: 0.82–1.33) after adjusting for all the determinants. As found in the separately urban and rural data, in the combined data diabetes, overweight, central obesity and insulin resistance were significantly associated with NAFLD. Additionally, we found that raised triglyceride levels (OR: 1.30; 95%CI: 1.01–1.67) became a significant determinant in the combined analysis (S2 Table).

## Discussion

In this population-based survey among a representative sample of adults between 30 to 60 years in North India, we found that three-fifths of the population had ultrasound detected NAFLD. We found a strong association between metabolic syndrome and NAFLD in multivariate logistic models. The high prevalence of the burden of NAFLD should be a matter of concern due to the future risk of severe liver disease (NASH, fibrosis, cirrhosis, hepatocellular carcinoma, mortality).

Three community-based NAFLD prevalence studies based on ultrasound from urban Mumbai (2007), urban Chennai (2009), and rural Haryana (2010) had found the prevalence of 16.6%, 32.0%, and 30.7%, respectively [16–18]. The current study has reported a substantially higher prevalence in both urban and rural India. A recent population-based study among 2089 adults aged ≥25 years from Trivandrum, Kerala, had reported a comparable prevalence of NAFLD 55.2% and 43.4% in urban and rural populations, respectively [19]. Besides, our results confirm the strong association of NAFLD with other metabolic factors-central obesity, diabetes, and triglyceride, which is consistently established in earlier studies [11, 16, 17, 19, 29]. Additionally, other factors might explain the urban-rural differences, such as gut microbiota. The role of gut microbiota in the pathogenesis of NAFLD is still evolving [30]. Changes in gut-liver homeostases, such as breakdown of the gut barrier, portal transfer of bacterial endotoxin

(lipopolysaccharide) to the liver, changed bile acid profiles, and lower concentrations of short-chain fatty acids, all contribute to the development of NAFLD [31].

This study is the first population-based survey to report the prevalence of a severe spectrum of hepatic injury in the form of raised enzymes and fibrosis based on VCTE. In the 2010 study from rural West Bengal, VCTE and liver biopsy was done only on those with NAFLD diagnosed by ultrasound or CT scan with raised ALT levels (2.3% of the sample). They reported that 2.4% of people with NAFLD had cirrhosis, and 0.2% of the population had NAFLD associated cirrhosis. In our rural sample, cirrhosis based on TE was seen in 0.64% of NAFLD subjects with raised alanine transaminase and 0.48% of the general population.

We observed in our study that 12.7% and 13.6% of people with non-fatty liver disease had raised enzymes, and 2.7% and 10.6% of them had fibrosis in rural and urban population, respectively. These point out the non-NAFLD related injury to the liver in this population. This asymptomatic liver injury was seen more in an urban area than in the rural area. In a recently published study on 6083 subjects in and around Delhi, elevated ALT ($\geq$40 IU/L) was reported in 20.5% of the participants. The prevalence of elevated ALT significantly declined with age [32]. The decline in NAFLD with higher age could be due to survival bias or could be because, with age, the body's fat content decreases, and thus, the early stages of NAFLD could be reversible.

We found a non-significantly higher prevalence among women than men in urban areas. When stratified by grades, we found that the prevalence of NAFLD in women was predominantly of Grade 1 while men had higher grades of NAFLD.

While many studies have found a higher prevalence of NAFLD among men [33], few recent studies have found preponderance in women [34]. Further, the mean age of women in our cohort was 44.8 years; therefore the women in our cohort might have lower estrogen protection [33, 35].

We observed differences in the prevalence of NAFLD based on ultrasound and CAP scores. The CAP scores were higher in the urban as compared with the rural population. These differences can be explained by the fact that CAP scores are affected by multiple factors. A recent individual patient meta-analysis which included more than 2346 patients from 9 countries, reported body mass index, diabetes, aspartate aminotransferase level, and sex to influence CAP scores [36]. In our study the urban cohort had a higher BMI, higher proportion of male subjects, and diabetes. CAP values for the presence of steatosis vary from 215–288 dB/m across studies due to differences in demographics, and laboratory parameters of included patients [23, 36–41].

Our study found only a fair agreement between ultrasound and VCTE. The LSM did not correlate with NAFLD activity score, age, and sex in the West Bengal study but showed a positive correlation with higher stages of hepatic fibrosis (Spearman's rho, 0.55).

The Government of India recently proposed using NAFLD as a gateway or sentinel strategy to include chronic liver diseases in the National Programme for Prevention and Control of Cancer, Diabetes, Cardiovascular Diseases and Stroke (NPCDCS). The use of raised ALT levels or ultrasound is being considered as possible screening strategies for integration into the NCD screening of a population above 30 years [42]. Our study has used several criteria to assess NAFLD using ultrasound, VCTE and hepatic marker- ALT. We found a good correlation of higher grades ultrasound based NAFLD with more specific methods VCTE and ALT. These findings can inform the NPCDCS in deciding the screening options for NAFLD at different health systems levels.

Our study has several strengths. We studied in a standardized fashion both urban and rural populations in a well-defined region using both sensitive and specific criteria such as measurement of serum hepatic enzymes and fibrosis by VCTE to establish the spectrum of

NAFLD in the representative adult population. All the ultrasound scans were read by a single radiology expert, our primary mode of diagnosis of NAFLD in the present study. Our analysis found a high degree of intra-observer agreement when the radiologist re-read 10% of blinded scans.

Our study has some limitations. One, liver biopsy is the gold standard for the diagnosis of NAFLD. However, since invasive diagnostic measures are inappropriate for a population-based study, ultrasound has been the mainstay for assessing hepatic steatosis. However, steatosis diagnosed based only on ultrasound is prone to misclassification. In our study, only one expert read all the scans and found reasonable intra-person reliability. However, it was difficult to differentiate between Grade 1 steatosis and normal liver scan, which may have led to misclassification. VCTE was done at two different centres by different technicians, which may have led to some subjective differences in measurement.

However, both the technicians were experienced, and measurements were standardised to minimise the possible difference. Two, the CAP values are affected by multiple variables, including body mass index, etiology of the liver disease, the extent of subcutaneous fat, and severity of hepatic steatosis. The prevalence of NAFLD would vary depending upon the CAP cut-off used for the grading of hepatic steatosis. Three, liver stiffness was based on LSM alone because APRI (AST to Platelet Ratio Index) and FIB-4 (Fibrosis-4) were not available. Four, autoimmune disorders, though their overall prevalence is very less, are also an important cause of fatty liver. We excluded those participants who had self-reported pre-existing liver disorders. However, it is hard to be certain about the cause of liver disease in community studies in LMICs like India, where the documentation of causes are not available. A probable reason for higher refusal in rural areas could be the travel distance between the villages and the nearest city where the evaluation (including USG and VCTE) was being performed. This study's generalizability across India is limited by the study sampling procedure, age group selected (30–60 years) and high refusal in the rural area. However, the prevalence of different metabolic syndrome components in our study is similar to those reported from other studies in North India [43, 44], suggesting that our observations are likely to be representative in a population with a high prevalence of obesity and diabetes.

In conclusion, we found a high prevalence of ultrasound-diagnosed NAFLD, NAFLD with raised alanine transaminase and fibrosis based on TE in both urban and rural populations in North India. In view of the high burden of asymptomatic NAFLD and its progression to irreversible forms of hepatic diseases and the resultant cardio-metabolic implications, there is a need for a public health approach, including screening and integration into the national NCD preventive and control measures.

## Supporting information

**S1 Table. Crude prevalence (%) of NAFLD based on different parameters in urban Delhi and rural Ballabhgarh.**
(DOCX)

**S2 Table. Univariate and multivariate analysis of factors associated with NAFLD defined on ultrasound in the combined dataset.**
(DOCX)

**S1 Questionnaire. NAFLD annexure1.**
(PDF)

## Acknowledgments

We acknowledge the contribution of the field team in data collection- Ms. Manju Sharma, Ms. Anita Yadav, Ms. Kajal, Ms. Minakshi Lal, Ms. Rukhsar, Mr. Virender Kumar, Ms. Anshu Arya, Mr. Tripurari Prasad Singh, Mr. Sanjay Kumar, Mr. Ram Narayan Singh, Mr. Pradeep Kumar, Ms. Varisha Khan, Mr. Hemchand Kandpal, Mr. Shailendra Kumar Singh, Mr. Lakhan Singh Rajput, data management- Mr. Naveen Kaushik, conducting ultrasound -Dr. Apoorv Goel, FibroScan technicians- Ms. Kavita Dhankar, Mr. Vinod Kumar and Mr. Babru Bhan Kaushik, Supervision of FibroScan at QRG- Dr. Sanjay Kumar. We also sincerely acknowledge Dr. S. K. Acharya, Former Professor and Head, Department of Gastroenterology, AIIMS, New Delhi for his initial guidance; Dr. Kavita Singh and Dr. Priti Gupta of Centre for Chronic Disease Control (CCDC) for their support.

## Author Contributions

**Conceptualization:** Roopa Shivashankar, Raju Sharma, Lakshmy Ramakrishnan, Dorairaj Prabhakaran, Anand Krishnan, Nikhil Tandon.

**Data curation:** Md Asadullah, Roopa Shivashankar, Devasenathipathy Kandasamy, Garima Rautela, Ritvik Amarchand.

**Formal analysis:** Md Asadullah, Dimple Kondal.

**Funding acquisition:** Nikhil Tandon.

**Methodology:** Roopa Shivashankar, Shalimar, Anand Krishnan, Nikhil Tandon.

**Project administration:** Md Asadullah, Roopa Shivashankar, Garima Rautela, Ariba Peerzada, Bhanvi Grover.

**Resources:** Dorairaj Prabhakaran, Nikhil Tandon.

**Supervision:** Md Asadullah, Roopa Shivashankar, Devasenathipathy Kandasamy, Garima Rautela, Baibaswata Nayak, Lakshmy Ramakrishnan, Nikhil Tandon.

**Validation:** Md Asadullah, Roopa Shivashankar, Shalimar, Devasenathipathy Kandasamy, Dimple Kondal, Anand Krishnan.

**Visualization:** Md Asadullah, Roopa Shivashankar.

**Writing – original draft:** Md Asadullah.

**Writing – review & editing:** Roopa Shivashankar, Shalimar, Devasenathipathy Kandasamy, Dimple Kondal, Garima Rautela, Ariba Peerzada, Bhanvi Grover, Ritvik Amarchand, Baibaswata Nayak, Raju Sharma, Lakshmy Ramakrishnan, Dorairaj Prabhakaran, Anand Krishnan, Nikhil Tandon.

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
