## [Decision Letter · Decision Letter 0]

27 Oct 2021

PONE-D-21-12623Rural-Urban differentials in prevalence, spectrum and determinants of Non-alcoholic Fatty Liver Disease in North Indian populationPLOS ONE

Dear Dr. Tandon,

Thank you for submitting your manuscript to PLOS ONE. After careful consideration, we feel that it has merit but does not fully meet PLOS ONE’s publication criteria as it currently stands. Therefore, we invite you to submit a revised version of the manuscript that addresses the points raised during the review process.

We look forward to receiving your revised manuscript.

Kind regards,

Daisuke Tokuhara

Academic Editor

PLOS ONE

Additional Editor Comments (if provided):

Thank you for giving an opportunity to review the valuable manuscript. The current manuscript is well designed and the discussion are well supported by the data, however several issues are raised by the reviewers. Please carefully address these points.

Journal Requirements:

4. Please provide additional details regarding participant consent. In the ethics statement in the Methods and online submission information, please ensure that you have specified whether consent was informed.

5. Please ensure that you include a title page within your main document. We do appreciate that you have a title page document uploaded as a separate file, however, as per our author guidelines (http://journals.plos.org/plosone/s/submission-guidelines#loc-title-page) we do require this to be part of the manuscript file itself and not uploaded separately.

7. Your ethics statement should only appear in the Methods section of your manuscript. If your ethics statement is written in any section besides the Methods, please move it to the Methods section and delete it from any other section. Please ensure that your ethics statement is included in your manuscript, as the ethics statement entered into the online submission form will not be published alongside your manuscript. 

Reviewers' comments:

Reviewer's Responses to Questions

**Comments to the Author**

1. Is the manuscript technically sound, and do the data support the conclusions?

Reviewer #1: Partly

Reviewer #2: Yes

2. Has the statistical analysis been performed appropriately and rigorously? 

Reviewer #1: N/A

Reviewer #2: Yes

3. Have the authors made all data underlying the findings in their manuscript fully available?

Reviewer #1: Yes

Reviewer #2: Yes

4. Is the manuscript presented in an intelligible fashion and written in standard English?

Reviewer #1: Yes

Reviewer #2: Yes

5. Review Comments to the Author

Reviewer #1: This cross-sectional study reports the prevalence of NAFLD and analyzes the associated risk factors in appropriate measure. Several limitations were disclosed sufficiently, such as the representative of study population.

1. Why do author select "aged 30-60 years" to instead of "aged 20-80 years" or "aged 20-60 years"? How is different "aged 20-30 years" ?

2. Are all patients with Diabetes type2?

3. Why don't you record the data about alcohol and autoimmune disease?

Reviewer #2: I deeply appreciate to have an opportunity reviewing this valuable research paper. The paper is well designed and written, but I have several minor comments to be addressed.

Comment 1; Authors should discuss CAP cut off value in the section of Discussion.

CAP is the important tool in this study for the screening of NAFLD. Authors used the cut off value (275dB/m) of CAP for determining the presence of NAFLD based on the previous study (Chalmers J et al. BMJ Open 2019). However there are currently no internationally established cut-off value of CAP of NAFLD. Numerous studies provided or used original cut off or reference value of CAP for NAFLD and healthy individuals in adult, children and adolescents. Results regarding the prevalence of NAFLD may be different according to the used cut off value of the CAP value. Therefore I strongly recommend to cite at least the following references from different countries in the revised manuscript. Then authors should add the study limitation in terms of the use of the cut off or reference value of CAP. For example, discrepancies of the cut off and/or reference value may relate to differences in the study design and populations including disease aetiologies, the prevalence of obesity and extent of subcutaneous adiposity, and the severity of steatosis, which may influence CAP performance.

(Reference 1) Sasso M, et al. Ultrasound Med Biol. 2010;36(11):1825-35.

(Reference 2) de Lédinghen V, et al. Liver Int. 2012;32(6):911-8.

(Reference 3) Tokuhara D, et al. Plos ONE. 2016;11:e0166683

(Reference 4) Isoura Y, et al. Obes Res Clin Pract. 2020;14(5):473-478

(Reference 5) Chon YE, et al. Liver Int. 2014;34(1):102-9.

Comment 2: Another limitation is that authors have assessed liver fibrosis by means of liver stiffness alone. There are several useful blood-biochemical fibrosis marker or indexes (e.g., hyaluronic acid, type-IV collagen, AST to platelet ratio index [APRI] or FIB-4 index) which are known to be useful in the screening of NAFLD. So, Authors are encouraged to use at least one of those blood-biochemical parameters to assess NASH in the revised manuscript. If there is no serum samples to be examined, authors should discuss the above points in the section of Discussion as the study limitation.

Comment 3: Generally male population are prone to develop NAFLD rather than female population. Some studies suggest that, in elderly population, female are prone to develop NAFLD because NAFLD-protective role of estrogen are impaired by the menopause. In the current study, prevalence of NAFLD is predominant in female rather than male population especially in the elderly. Please discuss any hypothesis in terms of the sex differences in the current result more deeply.

Recently NAFLD and NASH are known to be strongly associated with the gut microbiota. Is there any possibilities that rural and urban differences in prevalence of NAFLD is arised from gut microbiota differencies ? The latest review article (Tokuhara D. Frontiers in Nutrition. 2021;8:700058) well described the role of NAFLD/NASH. It is recommended to shortly discuss the potential involvement of gut microbiome in terms of the results in this study by citing the previous paper.

6. PLOS authors have the option to publish the peer review history of their article (what does this mean?). If published, this will include your full peer review and any attached files.

Reviewer #1: No

Reviewer #2: No

---

## [Author Response · Author response to Decision Letter 0]

10 Jan 2022

We thank the reviewers and editors for the thoughtful and valuable comments to improve the manuscript. We are now submitting the revised manuscript incorporating the changes suggested in the comments. We have uploaded our point-by-point response to the editor and reviewers’ comments.

---

## [Editor Report · Decision Letter 1]

27 Jan 2022

Rural-Urban differentials in prevalence, spectrum and determinants of Non-alcoholic Fatty Liver Disease in North Indian population

PONE-D-21-12623R1

Dear Dr. Tandon,

We’re pleased to inform you that your manuscript has been judged scientifically suitable for publication and will be formally accepted for publication once it meets all outstanding technical requirements.

Kind regards,

Daisuke Tokuhara

Academic Editor

PLOS ONE

Additional Editor Comments (optional):

I deeply appreciate authors' efforts to respond to the reviewers' comments. Revised manuscript are well organized with datasets. Thanks!
---

## [Editor Report · Acceptance letter]

2 Feb 2022

PONE-D-21-12623R1 

Rural-Urban differentials in prevalence, spectrum and determinants of Non-alcoholic Fatty Liver Disease in North Indian population 

Dear Dr. Tandon:

I'm pleased to inform you that your manuscript has been deemed suitable for publication in PLOS ONE. Congratulations! Your manuscript is now with our production department. 

Kind regards, 

on behalf of

Dr. Daisuke Tokuhara 

Academic Editor

PLOS ONE